# Adaptive QP algorithm for depth range prediction and encoding output in virtual reality video encoding process

**Hui Yang** *, **Qiuming Liu, Chao Song**

School of Software Engineering, Jiangxi University of Science and Technology, Nanchang, China

* yanghui13970929926@163.com

**Data Availability Statement:** All relevant data are within the manuscript and its Supporting Information files.

**Funding:** Natural Science Foundation of Jiangxi Province: Research on Caching and Transmission

## Abstract

In order to reduce the encoding complexity and stream size, improve the encoding performance and further improve the compression performance, the depth prediction partition encoding is studied in this paper. In terms of pattern selection strategy, optimization analysis is carried out based on fast strategic decision-making methods to ensure the comprehensiveness of data processing. In the design of adaptive strategies, different adaptive quantization parameter adjustment strategies are adopted for the equatorial and polar regions by considering the different levels of user attention in 360 degree virtual reality videos. The purpose is to achieve the optimal balance between distortion and stream size, thereby managing the output stream size while maintaining video quality. The results showed that this strategy achieved a maximum reduction of 2.92% in bit rate and an average reduction of 1.76%. The average coding time could be saved by 39.28%, and the average reconstruction quality was 0.043, with almost no quality loss detected by the audience. At the same time, the model demonstrated excellent performance in sequences of 4K, 6K, and 8K. The proposed deep partitioning adaptive strategy has significant improvements in video encoding quality and efficiency, which can improve encoding efficiency while ensuring video quality.

## Introduction

Advances in technology and the rapid development of digital media have made Virtual Reality (VR) an increasingly popular concept in modern society, allowing people to experience immersive 3D environments. The rise of VR technology has greatly enriched people's entertainment life, and has also shown great potential for application in industries such as education, healthcare, construction, and military [1–3]. To achieve a realistic VR experience, high-quality video content is essential. However, compared to traditional 2D videos, VR videos require higher resolution and larger data volume, which puts higher demands on transmission bandwidth and storage devices. In the encoding and transmission process of VR videos, how to effectively process and compress data to ensure the playback quality of the user end has become a key research issue. Depth range information, which refers to the distance between each pixel in the video and the camera, is an important reference data for VR video coding. The accuracy of its prediction is directly related to the quality of 3D scene reconstruction and viewing experience [4–6]. Effective depth range prediction (DRP) technology can enable

Strategy in content-centered wireless networks
with non-uniform features (Grant No.
20202BAB212003).

**Competing interests:** The authors have declared
that no competing interests exist.

encoders to better understand the 3D structure of the scene, optimize encoding strategies, compress irrelevant information, and retain important information for visual perception quality. In addition, the setting of adaptive Quantization Parameters (QP) for encoding output is particularly important in the VR video encoding process. QP determines the level of data compression during the encoding process, and a high QP value means a higher data compression rate, but it may also lead to more significant image quality loss [7–9]. It is crucial to study adaptive QP algorithms to achieve the optimal balance between bandwidth efficiency and image quality. In this context, the aim of this study is to explore DRP technology and adaptive QP algorithm for encoding output in VR video encoding, aiming to improve the viewing experience of end-users and promote the popularization and application of VR technology. Part 1 introduces the research objectives. Part 2 designs an adaptive depth partitioning strategy for VR video encoding. Part 3 tests the designed algorithm. Part 4 draws research conclusions.

## 1. Related works

In the field of VR video research, Chen HY et al. explored the block by block streaming transmission problem of 3D VR videos. Due to the high bitrate of 3D videos, traditional encoding methods resulted in wasted bandwidth resources. They proposed a novel adaptive streaming transmission method based on viewport prediction, which utilizes the scalability of efficient video encoding to achieve multi-level video quality, and optimizes adaptability using Cube-Map projection. This method outperformed traditional methods in terms of video quality and re-transmission delay when the bandwidth was low [10]. Huang Y et al. compared the effects of video and VR on enhancing teachers' interest and self-efficacy in managing the classroom, and studied the impact of VR simulation on 49 student teachers through a randomized pre—and post test experiment. Compared to watching videos, although it also brings more additional cognitive burden, VR simulation significantly enhanced the interest and self-efficacy of teachers and students in classroom management [11]. Chao Y P et al. examined the differences between 360˚ panoramic video and 2D video in medical students' learning of medical history collection and physical examination skills. 64 medical students were randomly assigned to watch two formats of instructional videos. The 360˚ video group significantly outperformed the 2D video group in terms of total skill score, physical examination, learner satisfaction, and cognitive burden. This indicates that panoramic videos provide an effective way to learn clinical skills [12]. Glaser N et al. investigated the potential application of VR video technology in assisting learning in individuals with autism. This study pointed out that although fully immersive VR systems pose challenges in practical applications, video-based VR is easier to develop and deploy compared to traditional VR systems. It could provide a safe and controllable learning environment for patients with autism. The study also discussed the current development trends in this field and the application of evidence-based practices for autistic learners, and explored the challenges in adoption and implementation, as well as possible directions for future research [13].

In the field of adaptive algorithm applications, Liu C et al. studied a deep learning based video compression filter—QA-Filter. This study proposed an adaptive mechanism called FSQAM, which combines frequency and spatial dimensions to optimize CNN filters, enabling them to more effectively handle different quantization noises. In addition, the combined use of frequency quantization and spatial quantization corporate adaptive mechanisms further enhanced performance. QA-Filter exhibited good video encoding performance under various QP conditions, effectively reducing the encoding bit rate. The standard test sequence "Basketball Drill" achieved a bit rate reduction of up to 9.16% on brightness signals [14]. Wang S et al. investigated a safety critical control strategy applicable to structurally unknown systems. It

utilized the control barrier function method, combined with dynamic regressor extension and mixing techniques, to provide a novel parameter identification rule. This control scheme not only ensured system safety during the identification process, but also quickly and accurately estimated unknown parameters, maintaining the robustness of the control system even in the presence of interference. The effectiveness of the algorithm was demonstrated through two simulation experiments, including the adaptive cruise control problem [15]. Si L et al. focused on the decision of coding unit (CU) division for internal encoding in diverse video coding standards. Considering the need to improve coding performance, the author designed a fast adaptive CU partitioning algorithm based on a joint random forest classifier to reduce the computational burden on the encoder. This algorithm had the ability to fairly compare all possible CU partition types and significantly reduced encoding time. Compared with standard encoders, it reduced encoding time by more than half while maintaining an acceptable increase in bit rate [16]. Xu Q et al. solved the problem of dual compression detection in the field of video forensics. A novel motion adaptive detection algorithm was proposed to address the challenges of dual compression and high motion displacement content with the same parameters. This algorithm could effectively identify intra frame prediction mode fluctuations and instability of prediction units (PU) that were not utilized during the first encoding. Compared with existing state-of-the-art methods, this algorithm exhibited higher performance and better robustness [17].

Recent studies have shown that VR video encoding and transmission technology is developing towards high efficiency and optimized user experience. The application of deep learning in the field of video compression is becoming an obvious trend, especially in the design of filters and parameter optimization in video compression. These studies emphasize the importance of maintaining video quality while reducing bit rates, and have made progress in effectively handling different quantization noises. This study focuses on DRP and adaptive QP algorithms in the process of VR video encoding, aiming to provide new optimization strategies for VR video compression and transmission, and improve user experience.

Compared with other studies, this study has innovation in the application of technology. Firstly, the encoding unit segmentation depth prediction method based on proximity depth correlation is applied in the model. This method quickly predicts the ideal segmentation depth of encoding units by evaluating image attributes, greatly reducing complexity. Secondly, in terms of mode selection strategy, the model will adaptively select modes in different video regions based on video content, which effectively improves encoding efficiency. Finally, in the adaptive QP algorithm, the weighted spherical peak signal-to-noise ratio weighting method is introduced for adaptive adjustment. It can adaptively adjust the viewing priority of different regions of spherical videos to improve image quality. Given the technical characteristics of the model, it can be applied to high-quality virtual reality spherical videos under large-scale data transmission. This model can reduce the amount of data transmission while ensuring image quality, thereby supporting large-scale real-time playback of virtual reality videos. In summary, this study achieved rapid prediction of encoding unit depth, adaptive mode selection, and weighted spherical peak signal-to-noise ratio weighted adaptive adjustment through innovative technologies and language applications, suitable for high-quality virtual reality spherical videos under large-scale data transmission, improving encoding efficiency and supporting real-time playback.

## 2. Design of depth partitioning adaptive strategy for VR video encoding

This study designed a comprehensive and efficient VR video encoding scheme with coherence. The first part is essentially a deep partitioning strategy. This method simplifies the

segmentation process of CU, reduces redundant operations, reduces coding complexity, and ensures the efficiency and quality of coding. The second part is the mode selection strategy. This strategy focuses on how to choose the appropriate encoding mode to optimize encoding efficiency while maintaining video quality. The third part is the design of an adaptive strategy, which intelligently adjusts the QP and balances the encoding efficiency and image quality based on the Weighted Spherical Peak Signal to Noise Ratio (WS-PSNR), effectively managing the size of the output stream. The deep partitioning strategy provides structural optimization for the encoding process. The pattern selection strategy ensures the selection of the most efficient encoding mode in the optimized structure during the encoding process. The adaptive strategy further optimizes data flow through QP adjustment while maintaining visual quality. Overall, these strategies work together to improve the efficiency of 360 degree video compression and reduce the necessary coding workload.

## 2.1 A depth prediction method for CU division based on neighborhood depth correlation

Depth range prediction is a technique used to optimize the virtual reality video encoding process, with the main purpose of improving encoding efficiency and reducing computational complexity by predicting the optimal segmentation depth of the coding unit (CU) in advance. This study achieved self-adaptive depth range prediction through four steps, including adjacent block depth analysis, cost function definition, depth prediction strategy design, and simplified segmentation process. Analyze the depth information of adjacent blocks through depth range prediction, predict the segmentation depth of the target CU in advance, simplify the encoding process, and reduce computational complexity.

Traditional VR video encoding requires splitting and subdividing the video encoding. This segmentation process is usually described as the fork of CU. Its purpose is to find the most suitable splitting depth for each Coding Tree Unit (CTU) in order to better implement coding optimization strategies [18–20]. The fork process requires traversing all possible segmentation paths of CTU, which continuously increases the complexity of coding work [21–23]. To this end, an innovative strategy has been proposed that can use deep category prediction to accelerate the differentiation process of CU. It quickly estimates the ideal segmentation depth of CU by evaluating the attributes of the image being processed, effectively simplifying redundant segmentation operations and reducing coding complexity, as shown in Formula (1).

$$RD\text{cost} = SSE_d + \lambda \cdot B_d \tag{1}$$

In Formula (1), $B_d$ is the number of bits in the partition. $d$ is the unit depth. $\lambda$ is the Lagrangian operator. $SSE_d$ is the sum of squared errors between the current image and the reconstructed image, calculated using Formula (2).

$$SSE_d = \sum_{i,j} Diff(i,j)^2 \tag{2}$$

The calculation method for $Diff(i,j)$ in Formula (2) is Formula (3).

$$Diff(i,j) = I_o(i,j) - I_R(i,j) \tag{3}$$

In Formula (3), $I_o(i,j)$ represents the original pixel value and $I_R(i,j)$ represents the reconstructed pixel value. The main idea of the design strategy is to estimate the potential segmentation depth of the target block by examining the correlation of depth indicators of adjacent blocks. The CU segmentation process is Fig 1.

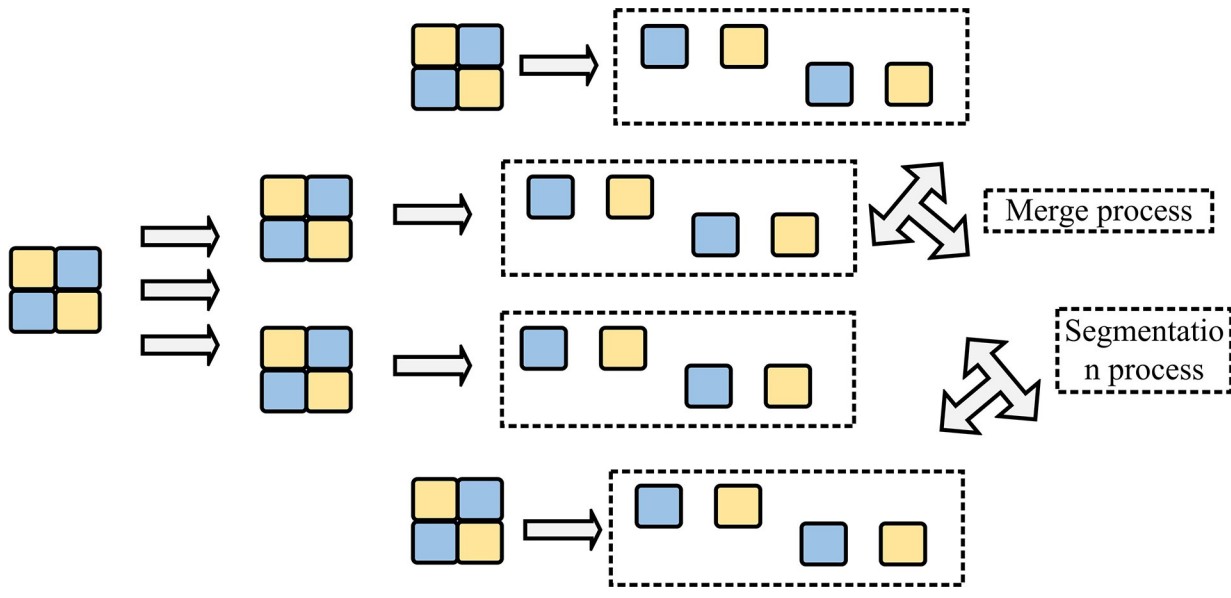

**Fig 1. CU segmentation process.**

In the ERP mode of imaging video clips, the polar regions are simplified in texture and increase in data repetition rate due to horizontal stretching, while the regions near the equator are less affected due to weak stretching. The division of CTU in the polar region is Fig 2.

Based on this observation, this technique has designed differentiated depth prediction strategies for different texture regions. In the polar region, horizontal stretching enhances the depth correlation between the adjacent blocks on the left and the processed blocks, and at this point, its weight is relatively large. The division of CTU in the equatorial region is Fig 3.

This study defines a novel cost function, namely the CU depth error, as a new evaluation criterion for the complexity of CU textures. Its calculation depends on the segmentation depth of surrounding blocks and the texture characteristics of the target block, as shown in Formula

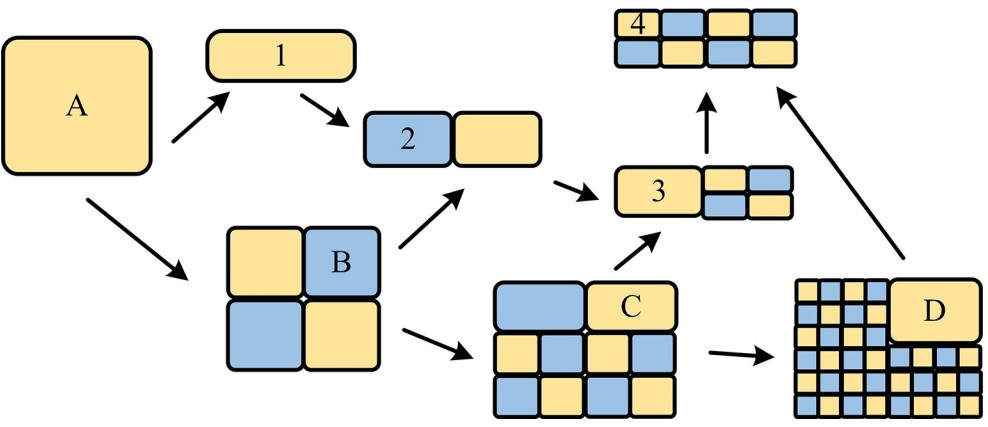

**Fig 2. The division of CTU in polar regions.**

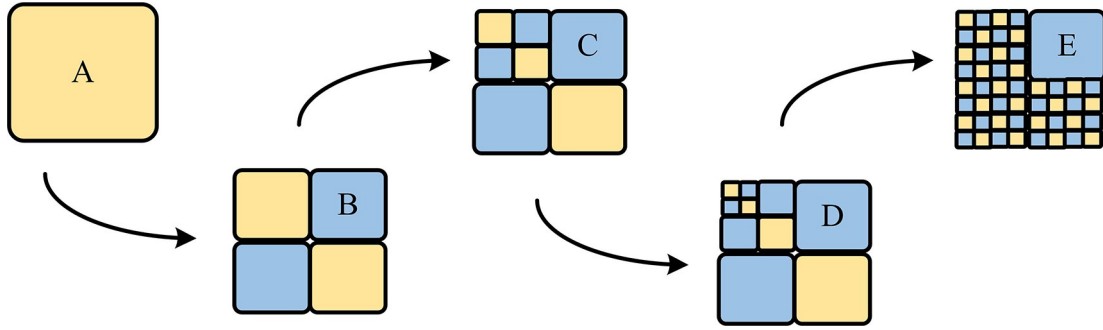

**Fig 3. The division of CTU in the equatorial region.**

(4).

$$CUD = \frac{1}{N}\sum_{i=0}^{N-1}|depth_i - d'|$$ (4)

In Formula (4), $d'$ represents the reference depth and is usually divided into four numerical cases, namely 0, 1, 2, and 3. $N$ represents the number of blocks, and $depth_i$ represents the depth value corresponding to the blocks. The depth division is Fig 4.

This study distinguishes blocks into adjacent blocks by calculating the CU depth error of other blocks adjacent to the current encoding block. It includes two types: simple segmentation and complex segmentation, and based on this, the appropriate segmentation depth of the target block can be inferred. In practical applications, it is necessary to first measure the depth error of all CUs within each CTU, classify CUs based on these values, and then determine the optimal depth interval of CUs by adding the set threshold to these classifications. This can achieve the effect of reducing coding complexity while maintaining coding quality.

## 2.2 A video encoding mode selection strategy using RDcost

The algorithm used to accelerate the encoding and decoding process focuses on improving processing speed and maintaining the quality level of the output video. This algorithm is

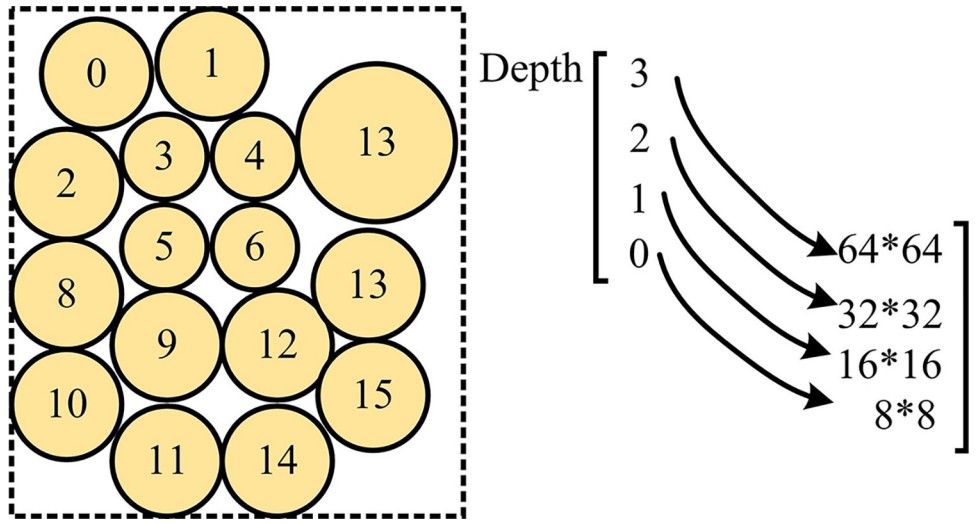

**Fig 4. Depth division situation.**

rooted in efficient video coding standards and implements precise strategy selection for CU to simplify the complexity of encoding and decoding. This can effectively reduce the necessary processing time and hardware resource usage. The core factor in determining how to segment video blocks is the calculated Quantification Rate Distortion Cost (RDcost). Before calculating this value, it is necessary to first compare the Sum of Absolute Transformation Difference (SATD) under all possible candidate configurations, which serves as a preliminary evaluation of distortion. Subsequently, based on the preset angle prediction mode, the video blocks are encoded and the configuration that minimizes SATD is selected. When searching for a suitable angle prediction configuration, it is necessary to calculate the corresponding SATD value by removing the residual signals of all angle modes except for some basic modes. After evaluating various RD costs, this is used to select the final encoding mode to be adopted, with special consideration given to the surrounding encoded video blocks. To achieve the goal of reducing coding complexity, adjustments can be made to specific regions by adaptively reducing the number of initial rounds selected in the selected pattern set. Specific regions, such as the polar and equatorial regions, will adjust their angle patterns based on the content of their respective coding blocks [24–26]. For example, in polar regions, due to distorted visual information, more emphasis will be placed on selecting patterns with more horizontal angles. The main difference when dealing with polar and equatorial regions is that the encoding of polar video content is relatively simplified, as its contribution to overall video quality is limited. The equatorial zone has a greater impact on the visual effect of videos, and requires higher details during encoding [27–29]. The selection strategy process is Fig 5.

For polar regions, the algorithm sets the spacing for mode selection based on PU depth and adjusts it according to the lateral stretching characteristics of the video, thereby improving the probability of selecting the best prediction mode (often horizontal mode). The equatorial zone retains the original selection interval to review the primary selection mode. Through this constant interval, the pattern that best reflects the content of the block is efficiently filtered out. The advanced mode decision stage further reduces the candidate list selected in the initial round, and pays special attention to the RDO cost of the current list and neighboring modes to select the final mode.

## 2.3 QP adaptive adjustment mechanism under WS-PSNR control

This study mainly utilizes the WS-PSNR weighting method to intelligently adjust QP to improve the efficiency of encoding processing, and effectively manage the output stream size while preserving the overall video quality. This algorithm can significantly reduce the load carried by data transmission. Firstly, it is necessary to deeply analyze the subtle aspects of quantization steps under efficient video coding standards, and then grasp the core advantages of utilizing rate distortion to optimize quantization. The quantization process aims to find an ideal balance between distortion and the necessary stream size, in order to achieve optimal rate distortion efficiency. This step is completed using the Lagrangian multiplier λ, which measures the relative importance of distortion and bit rate. In terms of 360 degree VR video compression, setting an appropriate QP is crucial for the final image quality and compression efficiency [30]. A small QP can ensure high-quality image output, but it also increases data traffic. On the contrary, a large QP will reduce data flow, but synchronously reduce image quality. Under equidistant cylindrical projection (ERP), the audience pays more attention to the image of the equatorial region, so assigning smaller QP to reduce distortion is beneficial. The distribution of projection weights is Fig 6.

Fig 6 shows the distribution of projection weights, where the darker the color, the closer it is to the upper and lower sides in the image, the closer its weight is to 0, and the lighter the

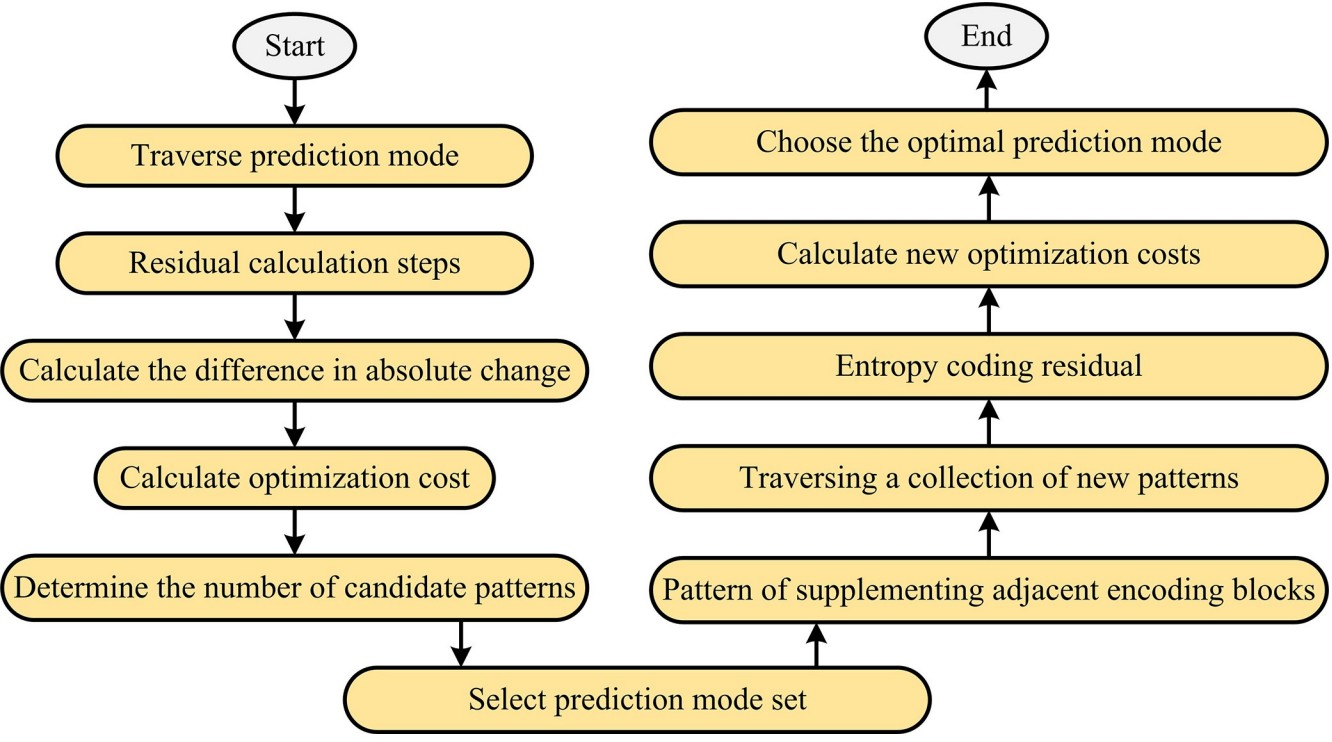

**Fig 5. Selection strategy process.**

color, the closer it is to the middle position in the image, the closer its weight is to 1. For bipolar regions that receive less attention, larger QP can be accepted in exchange for smaller bitstreams. In the encoding process, the weight factor considered in the WS-PSNR method can solve the non-uniformity of 360 video sampling and adjust the QP value reasonably based on this weight. The WS-PSNR calculation method is Formula (5).

$$WS - PSNR = 10 \log\left(\frac{MAX^2}{WMSE}\right) \tag{5}$$

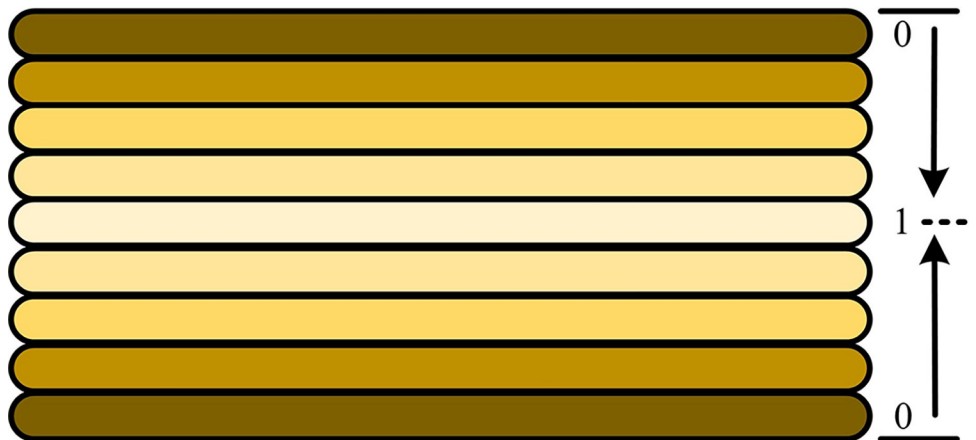

**Fig 6. Projection weight distribution.**

In Formula (5), *MAX* represents the maximum pixel value. The calculation of *WMSE* is Formula (6).

$$MWSE = \sum_{i=0}^{width-1}\sum_{j=0}^{height-1}(y(i,j) - y'(i,j))^2 \cdot W(i,j) \tag{6}$$

In Formula (6), $y(i,j)$ represents the original pixel value. $y'(i,j)$ represents reassembling pixel values. *width* represents the width of the video. *height* represents the height of the video. $W(i,j)$ represents the weight factor, as shown in Formula (7).

$$W(i,j) = \frac{w(i,j)}{\sum_{i=0}^{width-1}\sum_{j=0}^{height-1}w(i,j)} \tag{7}$$

In Formula (7), $w(i,j)$ represents the ERP weight factor, and the calculation method is as shown in Formula (8).

$$w(i,j) = \cos\left(\left(j - \frac{height}{2} + \frac{1}{2}\right) \cdot \frac{\pi}{height}\right) \tag{8}$$

This study further proposes a scheme for determining the QP correction values in different regions. After the previous exploration, reducing the average bit rate without sacrificing image quality, the QP in the equatorial region should be assigned a negative value correction. The bipolar region can tolerate higher QP to reduce the bit rate. The used update calculation is Formula (9).

$$QP_{new} = QP - 3 \cdot LOG_2(w) \tag{9}$$

In Formula (9), $3 \cdot LOG_2(w)$ represents binary logarithm, and coefficient 3 is an empirical parameter. The weight calculation method is Formula (10).

$$w(i,j) = \cos\left(\left(j - \frac{N}{2} + \frac{1}{2}\right) \cdot \frac{\pi}{N}\right) \tag{10}$$

The weight and average value of CTU for each individual row are shown in Formula (11).

$$w_{index-of-cttu-mean-weight}(i,j) = \frac{1}{N}\sum_{0+indexx\cdot N}^{N-1+index\cdot N} w(i,j) \tag{11}$$

In Formula (11), $index \cdot N$ represents the serial number, $N$ represents the CTU height, and $index-of-ctu-height$ represents the number. The average sum of all CTU weights is Formula (12).

$$Total\_ctu - mean\_weight = \sum_{0}^{num\_of\_ctu\_height} w_{index\_of\_cttu\_mean\_weight} \tag{12}$$

After adjusting the weight, Formula (13) is obtained.

$$w_{new} = \frac{w_{index\_of\_ctu\_mean\_weight}}{Total\_ctu\_mean\_weight} \cdot num\_of\_ctu\_height \tag{13}$$

The new update is Formula (14).

$$QP_{new} = QP - 3 \cdot LOG_2(w_{new}) \tag{14}$$

The update after compensation is Formula (15).

$$QP' = \min(51, QP - 3 \cdot \log_2(w)) \tag{15}$$

In the process of adaptive QP adjustment, it is necessary to first analyze the special issues of ERP projection video attributes and user focus areas, and clarify the necessity of adjusting QP. Using coding software to predict quantization levels for each transformation coefficient and select the most suitable quantization level according to RDO standards. Each CTU region uses WS-PSNR to determine weights and determine whether negative correction or positive correction is required. These weights are applied to the QP correction value calculation process and CTU encoding operations are performed. The entire encoder operation process requires continuous application of adaptive QP adjustment strategies and special handling of video content in different regions. When encoding, it is also necessary to ensure that the size of the adaptive QP compensated bitstream does not exceed the established extreme value, and WS-PSNR is applied to evaluate image quality. In the concluding stage, it is necessary to comprehensively evaluate the efficiency of the algorithm and video quality, and optimize the encoding efficiency based on test feedback. The entire design process should ensure that key factors such as WS-PSNR evaluation, audience attention to video content, and encoding efficiency are comprehensively considered to ensure the optimal and reasonable design of the algorithm. In addition, when designing algorithms, consideration should also be given to the limitations of coding tools and VR characteristics to ensure maximum improvement in coding efficiency and quality. The adjustment process is Fig 7.

The practicality and scalability of algorithms are also crucial in practical applications. The compensation mechanism needs to be compatible with current coding tools and can be validated through multiple test sequences to confirm its effectiveness in a wide range of application contexts. By comprehensively analyzing the WS-PSNR weights and adjusting the QP

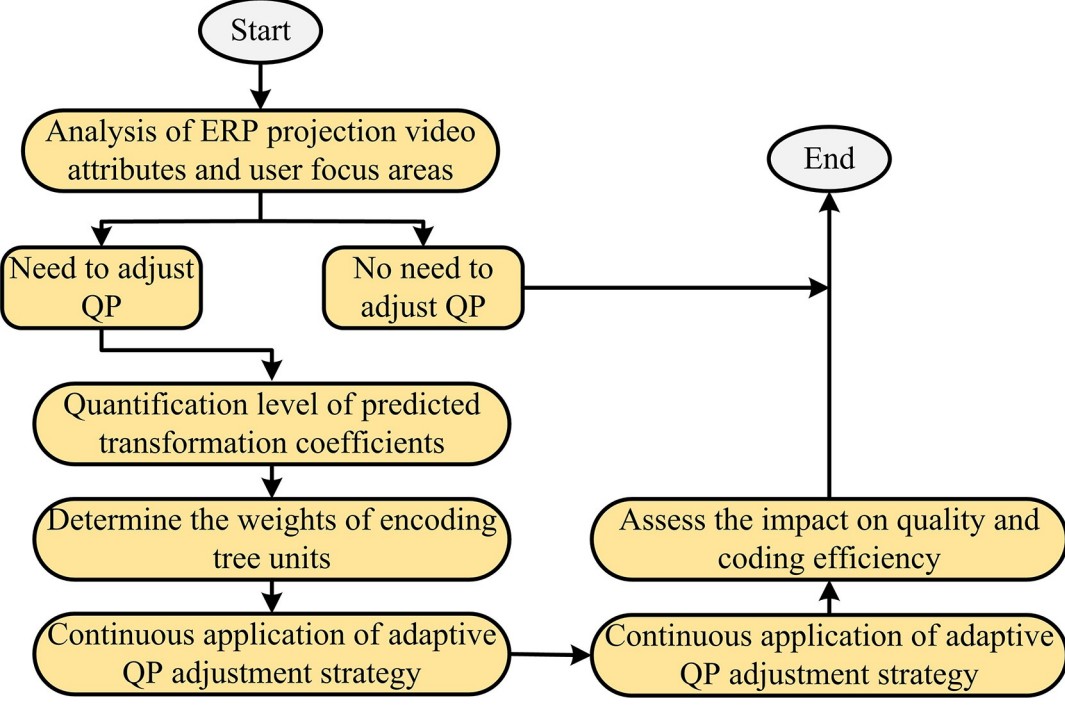

**Fig 7. Adjusting the process.**

values reasonably, this method has the ability to effectively reduce the required bitstream size for VR videos while ensuring image quality.

## 3. Verification of the effectiveness of deep partitioning adaptive strategy

This study tests the deep partitioning strategy and adaptive strategy separately. The main purpose of the former is to reduce coding complexity, while the latter improves coding efficiency through adaptive adjustment, and is analyzed in three scenarios: video resolution 4K, 6K, and 8K in the experiment.

### 3.1 Verification of the effectiveness of deep partitioning strategies

To verify the effectiveness of the proposed algorithm, a series of tests were conducted on a specially built experimental platform. The specific experimental settings are as follows. Hardware configuration is equipped with the computer of Intel® Core™ i7-7700 CPU and 8.0 GB of memory. The algorithm integration test used HEVC reference software HM 16.16 and Joint Vuleo Experts Team (JVET) strategy 360Lib-4.0. The experimental process mainly considered the All IntraMain (AI Main) encoding mode, and encoding was performed under QPQP of 22, 27, 32, and 37, respectively. The encoding frame rate is uniformly set to 100 frames. Table 1 shows the specific parameter settings.

In Table 1, performance analysis is mainly based on BRRR and time variation $\Delta T$ and WS-PSNR are three key indicators. BRRR, as an indicator for evaluating encoding efficiency, compared the rate changes of algorithms before and after implementation under the same image quality. Time variation $\Delta T$ is another indicator for measuring coding time savings. If the experiment shows a 15% reduction in coding time, it indicates that the algorithm significantly optimizes the coding process and improves the efficiency of the coding system. This is a crucial improvement for real-time video processing and video software in resource constrained environments. The final evaluation metric is WS-PSNR, which evaluates the quality difference between compressed and original videos. The decrease in WS-PSNR in the experiment is less than 0.1dB, which almost means that the audience is hardly aware of any quality loss, ensuring video quality while reflecting the high fidelity of algorithm compression. When evaluating the performance of video coding algorithms, the BRRR curve is a very important indicator. It can quantitatively represent how much bit rate one encoding algorithm saves compared to another encoding algorithm under the same visual quality. Based on the analysis of BRRR, this study can provide a visual impression of algorithm performance from the

**Table 1. Experimental setup.**

| Pilot projects | Setting Details |
|---|---|
| Encoding software | HM 16.16, 360Lib-4.0 |
| Hardware Platform | Intel® Core™ i7-7700 CPU @ 3.60GHz, 8.0 GB |
| Encoding Mode | All IntraMain (AI Main) |
| Encoding frame rate | 100 frames |
| Quantitative Parameters (QP) | 22, 27, 32, 37 |
| Performance evaluation indicators | Bit rate reduction rate (BRRR) (comparing bit rate changes under the same image quality) |
| | $\Delta T$ (Time variation, used to compare the degree of time savings) |
| | Weighted peak signal-to-noise ratio (determining the quality loss of reconstructed and original images) |

**Table 2. Minimal data set definition.**

| Dataset name | Resolution ratio | Sequence Name | Number of frames | Duration | Types | Describe |
|---|---|---|---|---|---|---|
| Dataset 1 | 4K | Urban scenery | 100 | 4s | Natural views | Expressing dynamic elements in urban landscapes, used to test the performance of coding in high detail scenes. |
| Dataset 2 | 6K | Natural scene | 100 | 4s | Natural views | Display natural landscapes and test the efficiency of coding strategies in handling high-resolution natural environments. |
| Dataset 3 | 8K | CityNights | 100 | 4s | City Night Scenery | Used to evaluate encoding performance in low light and high dynamic range scenarios. |

position and shape of the curve. The minimal data set definition used in the experiment is shown in Table 2.

This dataset is designed specifically for testing and evaluating the efficiency of video encoding. The dataset consists of video sequences with three resolutions: 4K, 6K, and 8K. Each sequence contains multiple application scenarios such as urban dynamic scenery, natural scenery, and urban night scenes, and includes effects from various aspects such as environmental dynamic details and ambient lighting. The comparison results between the model optimized by the deep partitioning strategy and the basic model are shown in Fig 8.

In Fig 8, the four video sequences in sub-graphs (a), (b), (c), and (d) all exhibit consistent trends. On the encoding efficiency curve, the BRRR of each sequence shows an upward trend with the improvement of quality. More specifically, these upward curves all showed rapid growth in the early stages. In the later stage, as the quality level improves, the growth rate of BRRR for each sequence gradually slows down. The emergence of this pattern reveals several important encoding characteristics. Firstly, the rapid increase in BRRR in the early stage indicates that within the low quality threshold range, a slight improvement in quality requires a significant increase in bit rate. This may be due to the lower quality of the initial encoding, so even a small improvement in quality requires a larger amount of data to achieve better visual effects. This is a noteworthy phenomenon for application scenarios that are sensitive to encoding efficiency. As the quality level improves, the slow rise of the BRRR curve in the later stage may mean that as the quality further improves, the required increase in bit rate is less. From the direction of the enlarged slope line, it can be seen that the slope of the later line is significantly smaller, indicating a significant decrease in the growth rate of the indicator. This shows that the encoder has good data compression efficiency under high-quality threshold conditions. This trend usually means that for users who pursue higher picture quality output, better picture quality effects can be achieved without significantly increasing transmission costs. From the consistent performance of these four sequences, it can be inferred that the research algorithm has good encoding efficiency, especially in the high-quality stage. The phenomenon of almost overlapping with the curve of JVET further emphasizes this point. Therefore, for the purpose of maintaining video quality, compared to JVET, research algorithms can provide similar or even more optimized coding efficiency at various quality levels, demonstrating their competitiveness in the field of coding.

This study used VVC software. As a new video encoding technology, VVC has more advantages in image quality compared to traditional technologies such as H.264/AVC and H.265/HEVC. It can compress at least 40% of the data while maintaining the same image quality. However, VVC has a limitation that it requires high encoding and decoding resources when dealing with complex video sequences or scenes. This leads to a decrease in the application efficiency of the model under resource constraints. However, from Fig 8, it can be seen that the research designed method can provide better encoding efficiency while maintaining the same image quality. As the quality continues to improve, it shows a late curve in Fig 8. At this time,

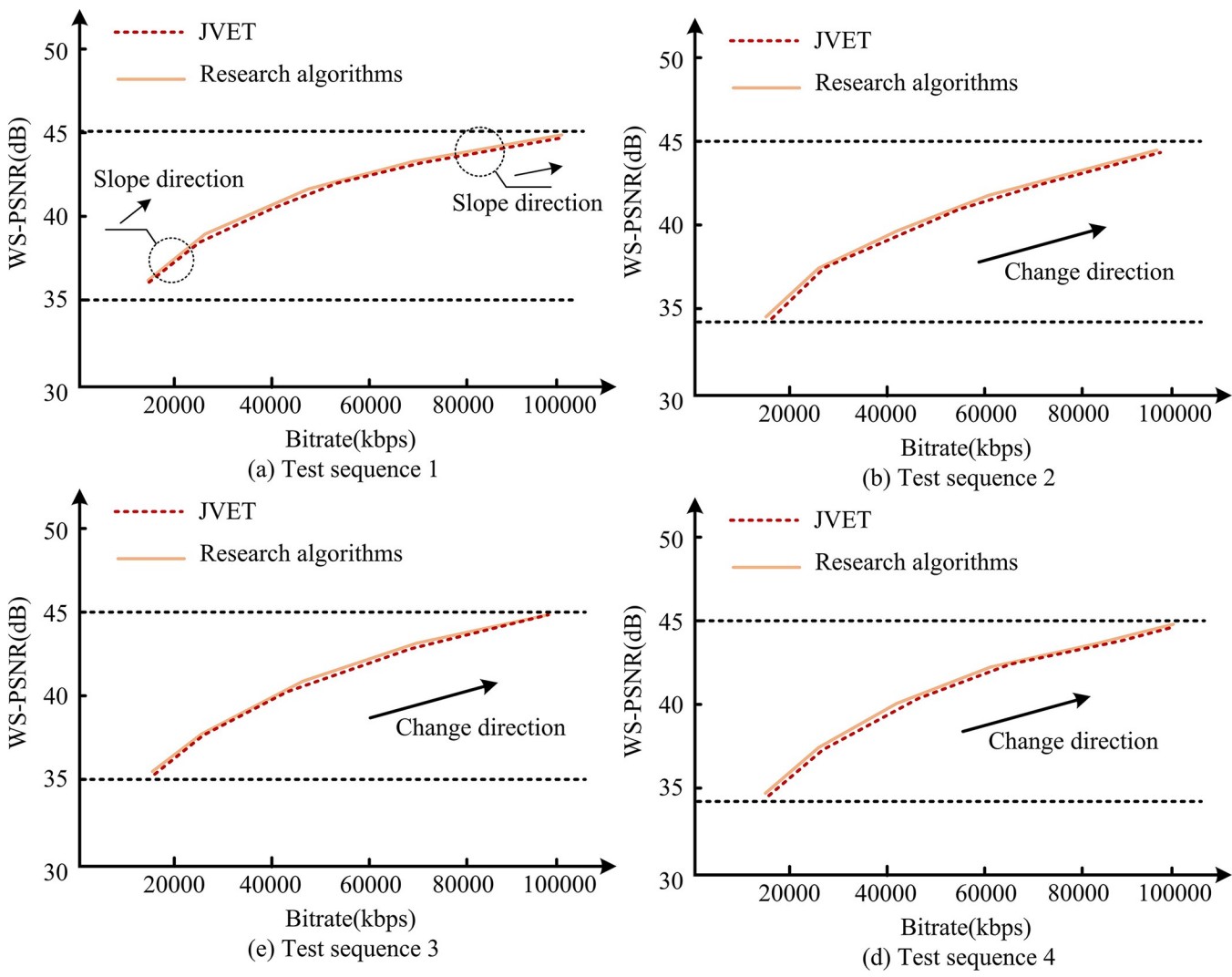

**Fig 8. Comparison of deep partitioning models.**

as the range of quality fluctuations narrows, the bit rate growth rate of each sequence gradually slows down, indicating that the model can achieve high-quality results with smaller bit rate increments. It can be seen that it has efficiency advantages in processing high-quality videos. It should be noted that JVET software is an open-source video encoding technology, and both VVC and JVET are standards maintained by the joint video encoding expert group of ITU-T and ISO/IEC. The division accuracy is Fig 9.

This study defines a novel cost function, namely the CU depth error, as a new evaluation criterion for the complexity of CU textures. Its calculation depends on the segmentation depth of surrounding blocks and the texture characteristics of the target block.Depth represents the reference depth and is usually divided into four numerical cases, namely 0, 1, 2, and 3. represents the number of blocks, and represents the depth value corresponding to the blocks. The depth division is Fig 4.

Fig 9(A)–9(C) represent the output values expressed by different data themes. Among all data topics, the accuracy of Depth 0 is generally distributed between 80% and 95%, the accuracy of Depth 1 is generally distributed between 80% and 90%, the accuracy of Depth 2 is

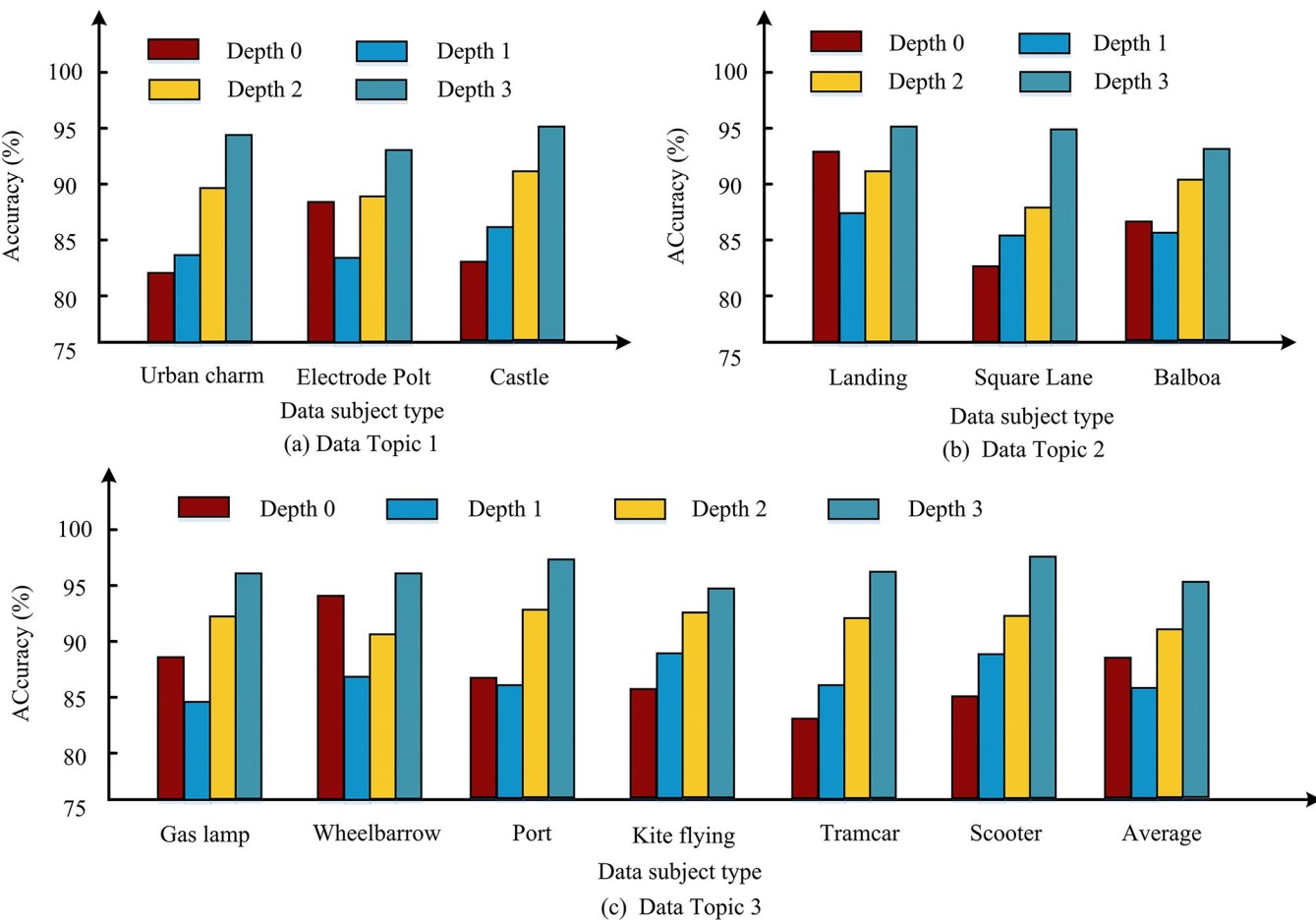

**Fig 9. Division accuracy.**

generally distributed around 90%, and the accuracy of Depth 3 is generally distributed around 95%. Overall, the accuracy of Depth 0 and Depth 1 is relatively similar, with Depth 3 having the highest level of accuracy. From this, it can be seen that the higher the Depth value, the more capable the model is in capturing information details. If we examine the CU division with a depth of 0 separately, the accuracy almost reaches the threshold of 90%. This achievement is crucial for high-level structural division, as errors at this level can be passed on to the next level. The CU partitioning with depths of 1 and 2 also demonstrated nearly 90% accuracy, reflecting the stability of the algorithm when dealing with medium sized CUs. At depth 3, although it involves the smallest unit size, the algorithm is still able to perform tasks accurately, indicating that it does not lose accuracy in capturing details. Therefore, based on a comprehensive balance, the research chooses a depth of 3 that can capture information details and has accuracy in executing tasks as the depth value for the application. Table 3 shows the indicator analysis of deep partitioning strategies.

In Table 3, the performance of our research algorithm in video compression encoding at different resolutions compared to the standard algorithm can be comprehensively analyzed by the variation values of three key indicators. These indicators include BRRR ΔT and WS-PSNR, positive values indicate an increase in indicators, while negative values indicate a decrease. Firstly, observing BRRR, the average value of each resolution is 0.83%, indicating that the

**Table 3. Index analysis of deep partitioning strategy.**

| Resolution ratio | Title | Bit rate reduction rate (%) | Time saving ratio (ΔT) | Weighted peak signal-to-noise ratio |
|---|---|---|---|---|
| 4K | Urban charm | 0.68% | 39.18% | 0.027 |
| | Electrode Polt | 0.68% | 38.54% | 0.048 |
| | Castle | 0.59% | 35.87% | 0.062 |
| | Landing | 1.08% | 42.65% | 0.035 |
| 6K | Square Lane | 0.98% | 39.18% | 0.038 |
| | Balboa | 0.78% | 42.75% | 0.038 |
| | Gas lamp | 1.38% | 41.15% | 0.041 |
| | Wheelbarrow | 0.48% | 41.76% | 0.048 |
| | Port | 1.08% | 41.63% | 0.043 |
| 8K | Kite flying | 0.68% | 34.10% | 0.058 |
| | Tramcar | 0.68% | 33.74% | 0.027 |
| | Scooter | 0.78% | 39.79% | 0.043 |
| Average | | 0.83% | 39.28% | 0.043 |

research algorithm has improved the bit rate by an average of 0.83% compared to the standard algorithm. For example, at a 4K resolution, the BRRR of "City Style", "Electrode Porter", and "Castle 2" are 0.68%, 0.68%, and 0.59%, respectively, all showing slight bit rate improvements. This means that researching algorithms has improved a small portion of data compression efficiency. In terms of coding time savings, ΔT shows a significant increase, with an average of 39.28% for each resolution. This positive value indicates that the research algorithm has reduced encoding time, thereby improving encoding efficiency. For example, at a resolution of 6K, the time savings for "Square Lane" were 39.18%, while for "Handcart" it was even higher at 41.76%, indicating that the algorithm significantly shortened the encoding time and improved the encoding speed. As an indicator of image quality, WS-PSNR shows a positive overall value, with an average of 0.043dB, especially reaching the highest value of 0.062dB in the "castle". This indicates that the research algorithm has completed the encoding work while maintaining or slightly improving image quality. Although the improvement was limited, coding time was saved without causing any quality loss. The division of UC is Fig 10.

In Fig 10, a comparison was made between the research model and Low Complexity Enhancement Video Coding (LCEVC) when analyzing the effectiveness of CU partitioning. The research method is more precise in division, and the comparison method shows situations such as missed division and multiple divisions.

## 3.2 Verification of adaptive models

This study first analyzed the ability of weight values in different image height intervals to adjust video encoding parameters. The weight values formed under different width windows vary as shown in Fig 11.

It is understandable how the weight values of different image height intervals in Fig 11 affect the adjustment of video encoding parameters, thereby affecting the final video quality. In the figure, within the first 800 width values of Height, WNew shows an upward trend. This means that the weight value gradually increases from the bottom of the video frame to the line with a Height of 800, reaching a maximum of 1.4W of the upper limit of change. This trend indicates that the visual importance of videos in this area is gradually increasing, and high accuracy needs to be assigned to this part during the encoding process to ensure video quality. As the Height value continues to increase, in the width range of 800–1600, WNew shows a downward trend. This indicates that within this region, as the image approaches the center, its visual importance decreases,

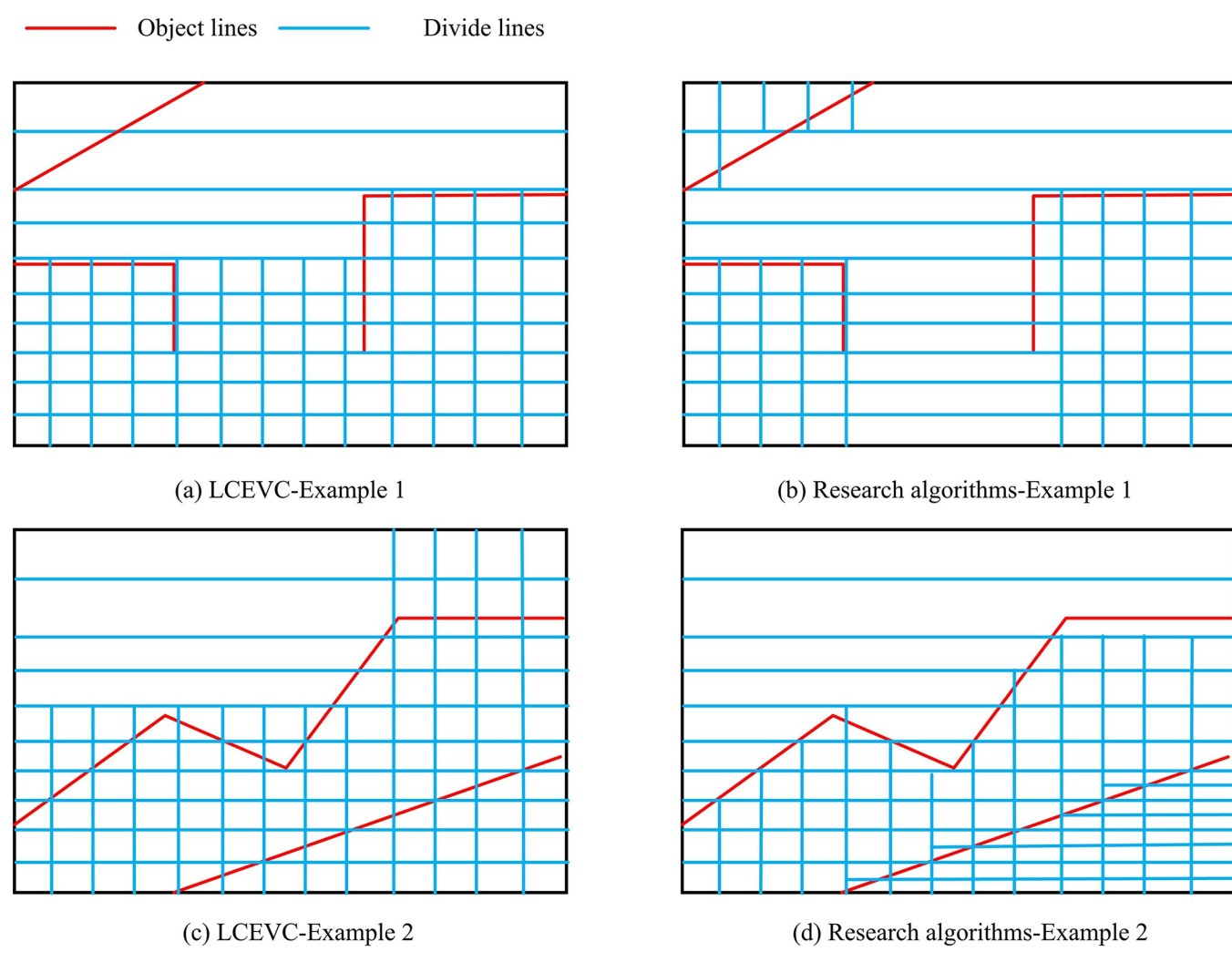

**Fig 10. UC division situation.**

and thus the reconstruction quality of these regions can be appropriately reduced during encoding to save bit rates. The analysis of the adaptive model BRRR is Fig 12.

Fig 12(A)–12(D) show consistent trends across the four video sequences. The comparison of research algorithms and coding frameworks for Versatile Video Coding (VVC) shows significant improvements. Taking the BRRR curve as a reference, the overall curve of the research algorithm is below the VVC curve. This means that under the same reconstructed video quality conditions, the bit rate required for researching algorithms is lower. Specifically, if the BRRR curve of the research algorithm shows a 5% lower than VVC at a certain quality point, it indicates that the algorithm can save 5% of data under the same image quality conditions. Table 4 shows the indicator analysis of the integrated adaptive model.

In Table 4, a comparison was made between the research algorithm and the basic methods, focusing mainly on two indicators: BRRR and WS-PSNR. Research algorithms exhibit varying degrees of differences in resolution and sequence. BRRR is an indicator of compression efficiency, with negative values indicating that research algorithms are more effective in data compression than basic methods. At a resolution of 4K, the BRRR difference of "Sky City" is -1.18%, indicating that the new algorithm reduces the data rate by 1.18% compared to the

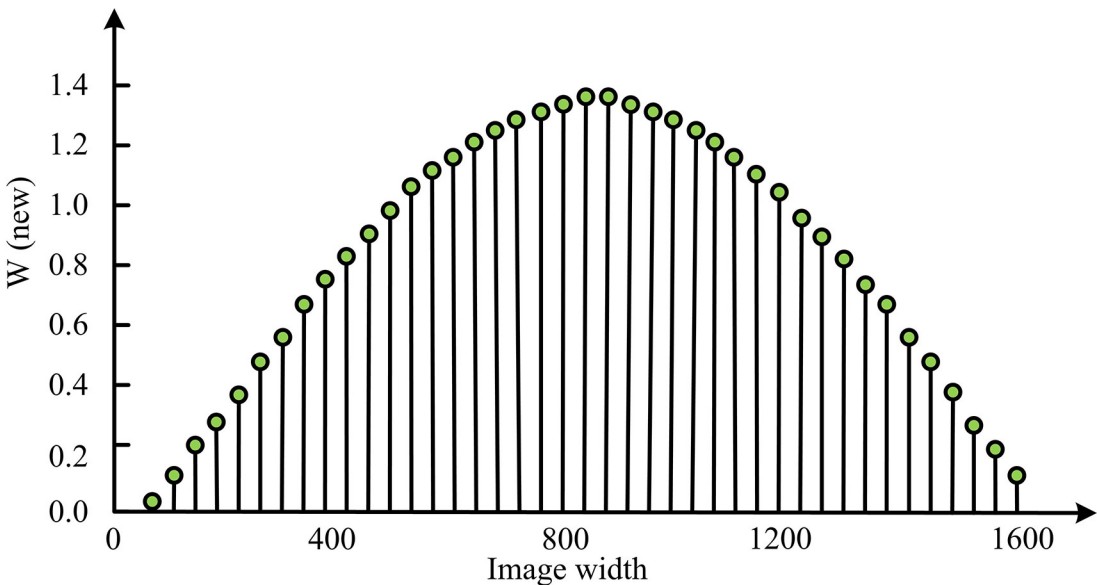

**Fig 11. Division accuracy.**

basic method. At 6K, the Broadway sequence showed a difference of -1.12%, demonstrating a certain improvement in compression efficiency. Overall, the average BRRR difference is -1.76%, indicating that the compression performance of the research algorithm generally exceeds that of the basic methods. WS-PSNR mainly evaluates image quality, which is an indicator to measure the difference in brightness quality between the reconstructed video and the original video. A positive value means that the research algorithm has improved visual quality compared to basic methods. Observing the data in the table, for example, on the 4K resolution pole vault sequence, WS-PSNR increased by 10.00%. At a resolution of 6K, the "Balboa" sequence showed the highest visual quality improvement, reaching 16.00%. Some sequences, such as the 6K descent 2, show a -1.00% change in WS-PSNR, indicating that in some cases, the new algorithm may be slightly lower in maintaining brightness quality than the base method. The average WS-PSNR change is 1.70%, indicating that the research algorithm has provided some visual quality improvement overall. Table 5 shows a horizontal comparison.

**Table 4. Indicator analysis of integrated models.**

| Resolution ratio | Sequence | Bit rate reduction rate | Weighted peak signal-to-noise ratio |
|---|---|---|---|
| 4K | Sky City | -1.18% | 8.00% |
| | Pole vault | -2.38% | 10.00% |
| | Braun Castle 2 | -1.87% | 8.00% |
| 6K | Landing 2 | -2.92% | -1.00% |
| | Broadway | -1.12% | 12.00% |
| | Balboa | -0.68% | 16.00% |
| | Gass Light Zone | -1.18% | 5.00% |
| | Tramcar | -1.99% | 6.00% |
| | Port | -2.18% | 4.00% |
| 8K | Kite flying | -1.97% | 4.00% |
| | Hanging Chair Journey | -1.69% | 6.00% |
| | Parking lot skateboard | -1.88% | 10.00% |
| Average value | | -1.76% | 1.70% |

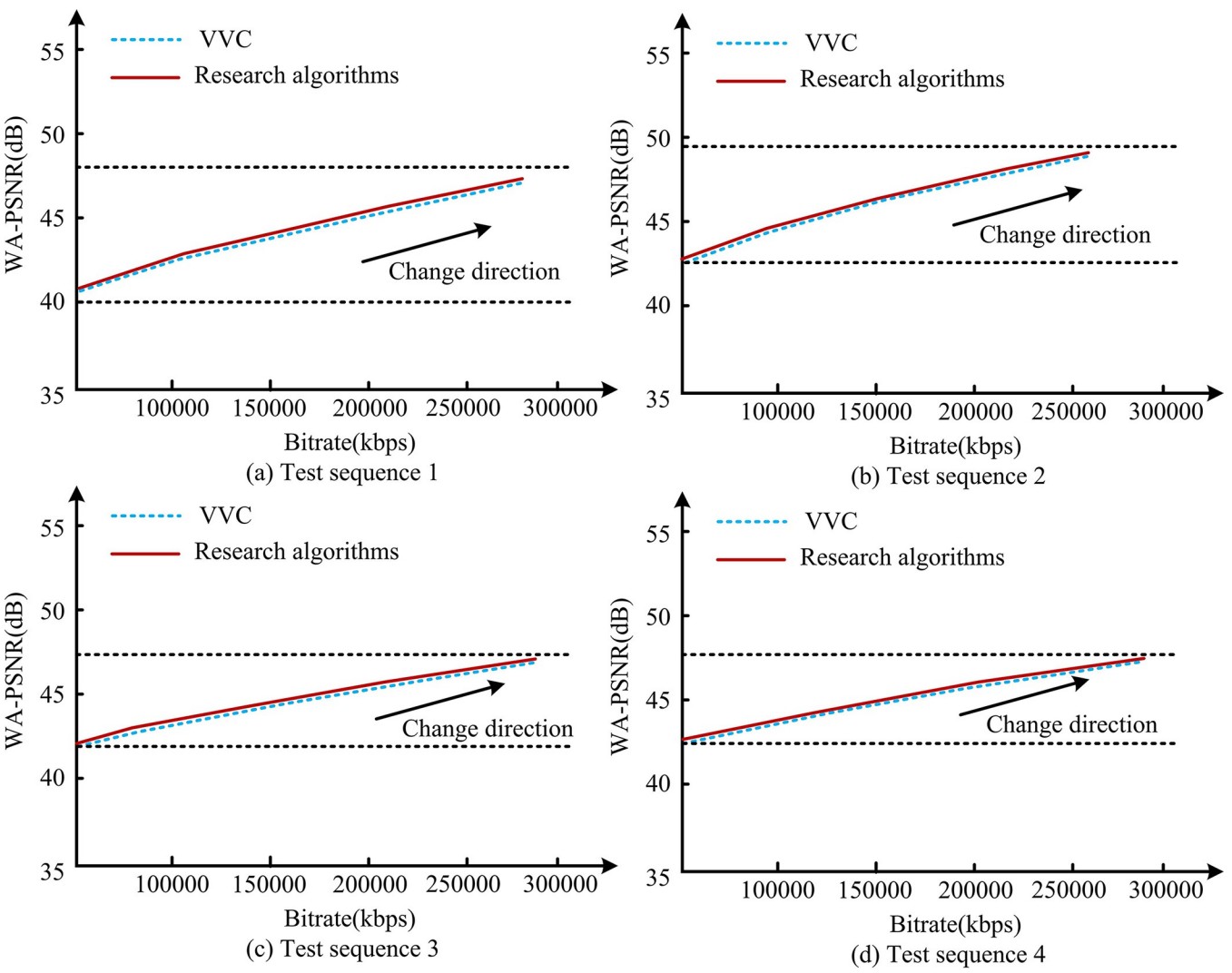

**Fig 12. Adaptive model BD rate analysis.**

Table 5 compares four models of Evolved Video Coding (EVC), Versatile Video Coding (VVC), AOMedia Video 1 (AV1), and Video Processing 9 (VP9) with the research model. In the quality fidelity stage, it is necessary to ensure efficient compression of video encoding while preserving the original visual quality. Comparison shows that VVC performs the best in this indicator, reaching 99.0%, while VP9 has the weakest performance, only 93.0%. In terms of coding efficiency, the research model also performed the most efficiently, with an accuracy

**Table 5. Horizontal comparison of models.**

| Model | Fidelity | Efficiency | Time Consumption |
|---|---|---|---|
| Research model | 98.50% | 92.00% | 1.2s |
| Evolved Video Coding | 95.00% | 89.00% | 1.8s |
| Versatile Video Coding | 99.00% | 93.00% | 1.5s |
| AOMedia Video 1 | 94.00% | 90.00% | 2.0s |
| Video Processing 9 | 93.00% | 87.00% | 2.2s |

of 93.0%. The lowest is VP9, with an encoding efficiency of 87.0%. The time consumption indicator measures the time required for the model to complete video encoding. The research model has the most obvious advantage in this regard, taking only 1.2 seconds to complete encoding, while other models take some more time, especially the VP9 model, which takes 2.2 seconds to complete encoding.

## Conclusion

This study proposed a deep partitioning adaptive strategy to solve the CU fast decision-making problem required for efficient video coding. This method fully considered the visual attributes of the video to predict and accelerate the partitioning process. At the same time, by intelligently adjusting QP, the encoding processing efficiency could be improved, and the output stream size could be managed while maintaining video quality. The results showed that in terms of division accuracy, the accuracy of the research algorithm was close to 90% in the four depth level divisions of 0, 1, 2, and 3. The coding efficiency saved an average of 39.28% of coding time. In terms of coding quality, the quality indicator value only decreased by 0.043dB, indicating that quality loss is almost imperceptible. In terms of bit rate, the research model even achieved a maximum decrease of 2.92%, with an average decrease of 1.76%. This strategy has also demonstrated its superior adaptability and effectiveness for sequences of different resolutions, namely 4K, 6K, and 8K. The average BRRR decreased by 0.83%, further reflecting the savings in the encoding process and the maintenance of quality. In summary, the research findings indicate that the proposed deep adaptive partitioning strategy achieves a good balance between quality and efficiency in video coding, and is suitable for future video coding standards and practical application scenarios.

## Supporting information

**S1 Data.**
(DOCX)

## Author Contributions

**Data curation:** Hui Yang, Chao Song.

**Software:** Qiuming Liu.

**Writing – original draft:** Hui Yang.

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
