## [Decision Letter · Decision Letter 0]

9 Feb 2024

PONE-D-24-01138Adaptive QP Algorithm for Depth Range Prediction and Encoding Output in Virtual Reality Video Encoding ProcessPLOS ONE

Dear Dr. Yang,

Thank you for submitting your manuscript to PLOS ONE. After careful consideration, we feel that it has merit but does not fully meet PLOS ONE’s publication criteria as it currently stands. Therefore, we invite you to submit a revised version of the manuscript that addresses the points raised during the review process.

We look forward to receiving your revised manuscript.

Kind regards,

Dr. Rahul Priyadarshi

Academic Editor

PLOS ONE

Journal Requirements:

3. We note that your Data Availability Statement is currently as follows: "All relevant data are within the manuscript and its Supporting Information files."

Reviewers' comments:

Reviewer's Responses to Questions

**Comments to the Author**

1. Is the manuscript technically sound, and do the data support the conclusions?

Reviewer #1: Yes

2. Has the statistical analysis been performed appropriately and rigorously? 

Reviewer #1: Yes

3. Have the authors made all data underlying the findings in their manuscript fully available?

Reviewer #1: Yes

4. Is the manuscript presented in an intelligible fashion and written in standard English?

Reviewer #1: Yes

5. Review Comments to the Author

Reviewer #1: This paper proposes an efficient way to encode 360-degree videos, using adaptive QP algorithm.

Polar-region aware QP adaptation algorithm is well known researches.

It seems that more novel method should be proposed for paper acceptance.

Of course, the proposed method showed gain compared to the anchor.

Experiments on EVC, VVC, AV1, and VP9 are convincing.

More explanations on Fig. 8 should be added, especially which software is used for JVET (is it VVC or exploration test model?).

6. PLOS authors have the option to publish the peer review history of their article (what does this mean?). If published, this will include your full peer review and any attached files.

Reviewer #1: No

---

## [Author Response · Author response to Decision Letter 0]

29 Mar 2024

The manuscript has been modified.

---

## [Decision Letter · Decision Letter 1]

2 Jul 2024

PONE-D-24-01138R1Adaptive QP Algorithm for Depth Range Prediction and Encoding Output in Virtual Reality Video Encoding ProcessPLOS ONE

Dear Dr. Yang,

Thank you for submitting your manuscript to PLOS ONE. After careful consideration, we feel that it has merit but does not fully meet PLOS ONE’s publication criteria as it currently stands. Therefore, we invite you to submit a revised version of the manuscript that addresses the points raised during the review process.

We look forward to receiving your revised manuscript.

Kind regards,

Zhaoqing Pan, Ph.D.

Academic Editor

PLOS ONE

Additional Editor Comments:

The authors need to carefully revise the paper according to the reviewers' comments.

Reviewers' comments:

Reviewer's Responses to Questions

**Comments to the Author**

1. If the authors have adequately addressed your comments raised in a previous round of review and you feel that this manuscript is now acceptable for publication, you may indicate that here to bypass the “Comments to the Author” section, enter your conflict of interest statement in the “Confidential to Editor” section, and submit your "Accept" recommendation.

Reviewer #1: All comments have been addressed

2. Is the manuscript technically sound, and do the data support the conclusions?

Reviewer #1: Partly

3. Has the statistical analysis been performed appropriately and rigorously? 

Reviewer #1: Yes

4. Have the authors made all data underlying the findings in their manuscript fully available?

Reviewer #1: (No Response)

5. Is the manuscript presented in an intelligible fashion and written in standard English?

Reviewer #1: (No Response)

6. Review Comments to the Author

Reviewer #1: 1. In Fig. 8, typo: WA-PSNR -> WS-PSNR.

2. In Related Works, what does it mean by "Language"? In the following sentence:

...Compared with other studies, this study has innovation in the application of

technology and language...

3. Depth range prediction (DRP) method is not well explained in the paper.

4. In the following sentence, "serial number" is not well explained.

...In formula (12), index represents the serial number, N...

5. In formula (10), why 3log2(w) was used? It would be better with explanation of this.

6. In Fig. 8, enlarged section of RD-curve needs to be added for the following sentence:

...As the quality level improves, the slow rise of the BRRR curve in the later stage may mean that as the quality further improves, the required increase in bit rate is less...

7. In Fig. 9, Part 1, 2, 3 are ambiguous.

8. In Fig. 9, regarding the depth, trade-off should exist between accuracy and encoding time for different depth values.

In later experiments, which depth value was used?

7. PLOS authors have the option to publish the peer review history of their article (what does this mean?). If published, this will include your full peer review and any attached files.

Reviewer #1: No

---

## [Author Response · Author response to Decision Letter 1]

12 Aug 2024

The manuscript has been modified according to comments.

Thank you!

---

## [Editor Report · Decision Letter 2]

9 Sep 2024

Adaptive QP Algorithm for Depth Range Prediction and Encoding Output in Virtual Reality Video Encoding Process

PONE-D-24-01138R2

Dear Dr. Yang,

We’re pleased to inform you that your manuscript has been judged scientifically suitable for publication and will be formally accepted for publication once it meets all outstanding technical requirements.

Kind regards,

Zhaoqing Pan, Ph.D.

Academic Editor

PLOS ONE

---

## [Editor Report · Acceptance letter]

16 Sep 2024

PONE-D-24-01138R2 

PLOS ONE

Dear Dr. Yang, 

I'm pleased to inform you that your manuscript has been deemed suitable for publication in PLOS ONE. Congratulations! Your manuscript is now being handed over to our production team.

Kind regards, 

on behalf of

Dr. Zhaoqing Pan 

Academic Editor

PLOS ONE